# The Relationship between Knowing Sign Language and Quality of Life among Italian People Who Are Deaf: A Cross-Sectional Study

**DOI:** 10.3390/healthcare11071021

**Published:** 2023-04-03

**Authors:** Sabina La Grutta, Marco Andrea Piombo, Vittoria Spicuzza, Martina Riolo, Irene Fanara, Elena Trombini, Federica Andrei, Maria Stella Epifanio

**Affiliations:** 1Department of Psychology, Educational Science and Human Movement, University of Palermo, 90123 Palermo, Italy; 2Department of Psychology “Renzo Canestrari”, Alma Mater Studiorum, University of Bologna, 40126 Bologna, Italy

**Keywords:** deafness, sign language, quality of life, social anxiety, self-esteem

## Abstract

Deafness is a medical condition with important relational implications. This condition could affect well-being and self-esteem and cause social anxiety. Sign language is not only a simple mimic but can be considered as a different kind of communication that could be protective for those who have learned it. However, some people do not use sign language because they think it can be marginalizing. The present study aimed to compare the quality of life (QoL) between people who learned Italian sign language as their first language with those who had never learned it or learned it later. This cross-sectional study involved 182 deaf Italian adults (70.3% females) who were recruited from Ente Nazionale Sordi (ENS) and by the main online deafness groups. The present results suggest that the deaf condition does not seem to significantly affect the dimensions of QoL pertaining to satisfaction and self-esteem, while it could have an effect on preventing high levels of social anxiety and in particular, the group who learned Italian sign language showed significantly less social anxiety than those who had never learned it.

## 1. Introduction

In a medical context, deafness is defined as a degree of hearing loss such that a person is unable to understand speech, even in the presence of amplification. However, deafness is not only a medical condition, but it also has important social implications because it affects not only the communicative dimensions of individuals, but also relational ones. Following the BIAP (Bureau International d’Audiophonologie) guidelines, different kinds of deafness can be categorized by severity grade (low, moderate, severe, and profound), injury (transmission, neurosensory, combined and central), or the time of language development (pre-verbal, peri-verbal and post-verbal).

In particular, children who have preverbal deafness were born with hearing impairment, and they need prosthetic support and rehabilitation therapy to develop language compared to their neurotypical peers. Children who have peri-verbal deafness develop hearing impairment around 3–4 years and lose the language previously acquired if they are not promptly prothesized. In contrast, people who suffer post-verbal deafness develop hearing impairment after complete language acquisition and preserve their linguistic heritage unaltered, but without proper implants, their verbal communication will not be adequately fluid and modulated, and this condition could affect the relational, social, and self-esteem dimensions [1].

In the developmental age, children with hearing impairment learn to speak in a dyadic relationship with their primary caregiver (often mothers) through repeated format (routines). With this repetitive behavior, mothers allow children to understand and predict their actions, thus becoming an active part of the interaction [2]. In this phase, oral language plays a fundamental role. Still, in mother–infant relationships, many other non-verbal messages are expressed by looking, body contact, and smiling, allowing communication beyond spoken language. The caregiver refers to situational context constantly, talking about things that can be seen and manipulated by the child and using pointing and gesticulation [3].

In this way, children with a hearing impairment can understand what is happening around them through the undamaged visual-gestural channel with natural and spontaneous communication with their caregiver [3].

### 1.1. Cultural and Social Dimensions of Deafness

Many people in the deaf community use sign language to communicate. They usually communicate with other people who are signers or receive assistance from an interpreter. This last option, in many cases, is the preferred one, even if they could have good language competence, thanks to lip reading and hearing residuals, to have better communicational access through their language [4]. This is because deafness is also considered as a socio-cultural condition for them. Some people do not want to use sign language but only lip read or hearing residuals [5] because these people think that sign language can be seen as marginalizing and consider deafness as a handicap [6].

In this regard, in recent years, deafness has been defined as a “deaf norm” [4,7]. In deaf studies [8], for example, the term “deaf”, with the first lowercase letter, has been opposed to “Deaf”, with the first capital letter, to describe two different ways of interpreting deafness and being deaf. They can be “audiology deaf” but not share “social facts” with other people who are deaf and trying to integrate themselves exclusively with hearing people, or they can be “signers deaf” and belong to the deaf community [5]. In this regard, from the English debate about deaf culture has emerged the term “deafhood” to underline linguistic and cultural identity in contrast to “deafness”, which refers to the audiology condition [4,5].

### 1.2. The Importance of Learning Sign Language

In the past years, in Italy, research about the communication of people who are deaf has highlighted how sign language is not only a simple mimic but can be considered as a different kind of communication that expresses itself through a visual–gesture modality instead of an acoustic–vocal one [9]. The “sign language” definition underlines that it can be considered as a language for all intents. Therefore, it has complex characteristics, specific symbol systems, and grammar rules shared by the deaf communities. In addition, it is important to differentiate between “sign”, which indicates the set of manual movements and/or facial expressions used by people who are deaf, and “gesture”, used by hearing people to complement verbal language [10]. There is no universal sign language, but it can be defined as “zoned” because each nation has different signs languages: American Sign Language (A.S.L.) or Langue des Signes Francaise (L.S.F.), British Sign Language (B.L.S.), and the Lingua Italiana dei Segni (L.I.S.). In Italy, deafness affects 1% of children [11]. About 95% of children with hearing impairment have hearing parents. However, among those with a parent who is deaf, only 5% of these parents use sign language as their first language. In this situation, only a few children with hearing impairment can learn signs spontaneously from the first months of life. Children who develop a hearing impairment later often cannot have the opportunity to learn both sign and spoken language naturally. Both of them are taught with an explicit education, usually late. Spoken language is acquired only after years of intense acoustic and speech therapy, and sign language is often taught only when other teaching systems fail [12].

In other cases, people who are deaf can decide to learn sign language in adulthood because they feel they belong more to the deaf community than the hearing one, from which they can feel excluded or have a more natural communication channel [13].

Offering different kinds of stimulations is fundamental because the language development of a child who is deaf, exposed to sign language from birth, has the same stages that a hearing child has in the acquisition of spoken language. At first, there is the “Manual Babbling” phase, then the holophrases phase, and later, by composing more songs, they can produce more articulated phrases [14,15]. Late exposure to oral language significantly affects language development in a subject with hearing impairment, which could be compromised. They cannot follow a steady developmental path in spoken language, with inevitable production and comprehension errors [3,16,17].

However, there is a chance for children with hearing impairments to combine the use of oral language, which they acquire by using a cochlear implant or hearing prosthesis, and sign language [18]. This bilingual and bimodal approach can promote children’s speech, communication, and cognitive development [19,20]. In this regard, a study by Jimenez et al. [21] on children with cochlear implants, comparing those who received only oral education with those with bilingual and bimodal education, showed that the latter obtained significantly better scores in verbal fluency. These results indicate that using sign language can promote access to oral language. The bimodal bilingual approach is important and necessary to permit people who are deaf to develop total language competence [13]. Sign language also has a good influence on reading ability [22,23], and as a coded transposition of oral language, it is also affected in people with hearing impairment.

Furthermore, sign language could promote satisfying relationships based on a standard communication system [24]. Regarding this, one study by Hintermair [25] examined the role of bilingual bimodal acculturation on self-esteem and life satisfaction; his findings showed that people who are deaf with marginal acculturation collectively have lower self-esteem and show less satisfaction with life than people with bimodal bilingual education. These results suggest how important it is for the promotion of mental health in people who are deaf to understand the factors related to their self-esteem. Self-esteem could be considered as an important resource for the quality of life, and it is associated with both reduced proneness to depression and a lower level of anxiety [26]. Regarding this, some studies have already highlighted the need for one’s psychosocial well-being to have a cultural anchor [27,28], supporting the idea that developing a healthy deaf identity and a home cultural identity requires a bimodal bilingual education [29,30,31]. Despite this evidence, there are still a few studies that have focused on the use of sign language and the quality of life of people who are deaf, and, to our knowledge, no studies have evaluated the effects of bimodal bilingual education on the dimensions of self-esteem and social anxiety in adulthood, and the impact of the precocity in which sign language was learned has on these dimensions.

The present study compares the quality of life between people who have learned ISL as their first language and those who have never or later learned it.

The hypotheses are:(1)Participants who learned ISL show low social anxiety and high life satisfaction and self-esteem than those who had never learned ISL.(2)Participants who learned ISL as a primary language showed better results in every QoL dimension such as self-esteem, social anxiety, and life satisfaction than those who later learned it.

## 2. Materials and Methods

### 2.1. Procedure

In this study, all questionnaires were administered online using Google Forms, and each item was also presented with a video translation in ISL. The translation was carried out by a researcher with a third-level ISL degree supported by the Sign Dictionary of Orazio Romeo and “The Italian Sign Language Dictionary, Spreadthesign”. Every item was transcribed following ISL grammar rules and later video-recorded. This study was conducted as per the Declaration of Helsinki and approved by the University of Palermo ethics committee (n.64/2021).

### 2.2. Participants

The present study is a cross-sectional design that involved 182 participants who are deaf (128 females; 70.3%; mean age = 37.21; SD = 10.31). Most (124; 68.8%) were born deaf, and only 38 (21%) had parents who were deaf. Finally, 128 of the participants (70.3%) knew ISL. Participants were recruited from Ente Nazionale Sordi of Trapani (Italy) and by online groups of people who are deaf. The inclusion criteria for the sample were: age (18–60) and the severity of their condition (only participants with profound deafness were recruited).

### 2.3. Measures

To measure quality of life, three different variables were evaluated: life satisfaction, social anxiety, and self-esteem.

#### 2.3.1. Demographics

Demographic data were collected using an ad hoc questionnaire to collect information about gender, age, if they had been deaf since birth or not, if participants had parents who were deaf or not, if the participants had learned ISL or not, and the age at which they had learned it.

#### 2.3.2. Life Satisfaction

Life satisfaction has two components: emotional and judgment (cognitive). In this study, the Satisfaction with Life Scale [32] was used to measure the cognitive component. It consists of 5 items in a 7-point Likert scale with a total score range of 5 to 35. A range of 5–9 indicates an extremely low life satisfaction, 10–14 indicates general unsatisfaction, 15–19 indicates scores slightly below the average, 20–24 indicates average scores, 25–29 indicates high satisfaction, and 31–35 indicates maximum life satisfaction.

#### 2.3.3. Social Anxiety

Social anxiety was measured using the Social Anxiety Scale (SAS-30) [33]. This scale is a 30 item scale that measures seven dimensions: Fear, Social Avoidance, Self-efficacy, Communication Difficulties, Being Watched, Dyadic Relationship with Strangers, and Demophobia [33]. The SAS-30 only provides a total score obtained by the sum of the values of each item. The total score was between 30 and 150, with higher scores corresponding to higher levels of social anxiety. In particular, the score ranges were: very low = 30–49; low = 40–56; average = 57–94; high = 95–114; very high = 115–150.

#### 2.3.4. Self-Esteem

Self–esteem was measured using the Italian version of the Rosenberg Self-Esteem Scale [34]. It consists of 10 items that measure the degree of the global self-esteem of individuals by expressing how much they agree or disagree with the statements along a 4-point Likert scale. Regarding the scores, the total is between 0 and 30; 15–25 is considered the average range, and scores below 15 indicate low self-esteem [35].

### 2.4. Data Analyses

Statistical analyses were performed using programs in the Statistical Package for Social Sciences (SPSS for Windows release 25.0). The significance of observed associations and differences between groups was tested using analysis of variance (ANOVA, F test) and the chi-square statistic (Pearson’s χ^2^). A difference was considered to be statistically significant with *p* < 0.05.

Once the entire sample was recruited, it was divided into three different groups, with no significant differences between gender and age composition (*p* > 0.05) following only the criteria of knowing ISL or not and, for those who learned ISL, the age at which they learned it. The first group was composed of 54 participants (41 females, 13 males) who had never learned Italian sign language (ISL); the second was composed of 71 participants (53 females, 18 males) who learned ISL before 18 years of age; the third was composed by 58 participants (34 females, 24 males) who learned ISL after 18 years old.

The group of those who learned ISL before 18 years old was also sub-divided into three different subgroups to evaluate the efficacy of preventive learning: the first was composed of 42 participants (32 females, 10 males) who learned ISL as their first language (between 0 and 5 years old); the second was composed of 16 participants (11 females, five males) who learned ISL between 6 and 11 years old; the last was composed of 13 participants (10 females, three males) who learned ISL later, between 12 and 17 years old.

Additionally, the scores of all variables were divided into three levels (high, average, and low) following the cutoff score indication of each test. The chi-squared test was used to verify between-group differences in each. In particular, comparisons were made between groups sorted by demographics (gender and age), by knowing ISL or not regardless of the learning period, and by the ISL learning period (0–5, 6–11, 12–17, and >18).

## 3. Results

### 3.1. Descriptive Results

The information about the composition of the three groups and the total sample are described in Table 1. With regard to the psychological variables, the overall mean scores of the sample were life satisfaction (M = 23.80; SD = 6.38); social anxiety (M = 83.99; SD = 19.30); self-esteem (M = 19.76; SD = 4.62). Regarding gender differences, the mean scores were similar in satisfaction (females: M = 23.86; SD = 6.16; males: M = 23.67; SD = 6.99) and self-esteem (females: M = 19.87; SD = 4.58; males: M = 19.48; SD = 4.73) with no significant differences (*p* > 0.05); females scored higher than males in social anxiety (females: M = 85.08; SD = 19.69; males: M = 81.41; SD = 18.25;) but not significantly (*F* = 1.37; *p* > 0.05).

With regard to age, three ranges were used (18–30; 31–50; >50,) and these groups showed no significant differences in all variables (*p* > 0.05) (see Table 2).

Regarding the knowledge of ISL, the mean scores in SWLS and RSES were very similar between the groups of those who learned ISL (after and before 18 years) and those who never learned it. This last group tended to show higher mean scores than the others in the SAS scores, even if the mean differences were not statistically significant (F = 0.56; *p * > 0.05) (see Table 3).

### 3.2. Life Satisfaction, Social Anxiety, and Self-Esteem Levels of the Sample

In order to verify the hypotheses of this study, we divided the scores of the psychological variables into three levels (low, average, and high) according to the cutoff ranges of each test. First, when examining the total sample, it showed mainly high levels of satisfaction (56.6%) followed by average levels (33%), and only 10.4% had low satisfaction levels. With regard to social anxiety, more than half of the total sample showed average levels of social anxiety (58.2%), while more than a quarter (33.5%) showed high scores, and only 8.2% showed low social anxiety scores. Finally, regarding the self-esteem levels, the total sample mainly showed average scores (74.7%), while high and low self-esteem levels represented only 12.6% of the sample, respectively (see Table 4).

### 3.3. Level Comparisons between Those Who Learned ISL and Those Who Not

To verify the first hypothesis, we compared those who had learned ISL on the whole (grouping together the two groups) (N = 128) and those who had not (N = 54). Both groups mostly showed high levels of satisfaction (No ISL = 53.7%; ISL = 57.8%), followed by average levels (No ISL = 33.3%; ISL = 32.8%) and low levels (No ISL = 13%; ISL = 9.4%), finding no significant differences between groups (χ^2^ =0.58; *p* > 0.05).

With regard to the social anxiety levels, the group of those who had never learned ISL showed 44.4% of high anxiety levels and only 11.2% of low scores, while the sample including those who learnt ISL mainly showed average scores (64.1%), followed by 28.9% of high scores, and only 7% of low scores, and these differences were statistically significant (χ^2^ = 6.01; *p <* 0.05).

Finally, regarding the self-esteem levels, both groups showed a great percentage of average scores (70.4% and 76.6%, respectively), similar results in high self-esteem (13% and 12.5%), while those who had never learned ISL showed the worst results in low self-esteem levels (16.7%) than those who had learnt it (10.9%), but these were not statistically significant (χ^2^ = 1.18; *p* > 0.05). See Table 4.

### 3.4. Level Comparisons between Those Who Learned ISL as a Primary Language and Those Who Learned It Later

To test the second hypothesis, only the participants who had learned ISL were considered in the analysis. Specifically, the group of those who learned ISL before 18 years of age was sub-divided into three sub-groups to evaluate if there were differences in the levels of psychological variables related to the age of learning: 0–5 years (as primary language); 6–11 years; 12–17.

Regarding the life satisfaction levels, the 6–11 group showed the highest percentage in high levels (66.7%), followed by 0–5 (59.5%), >18 (56.9%), and 12–17(46.2%), while the 12–17 group showed the highest percentage of low scores (23.1%) followed by 0–5 (9.5%), >18, and 12–17(6.9% and 6.7% respectively), but these differences were not statistically significant (χ^2^ = 4.35; *p* > 0.05).

Regarding social anxiety, the group that showed higher levels was 12–17 (46.2%), followed by 0–5 (28.6%), 6–11 (26.7%), and >18 (25.9%), while with regard to low levels, the 0–5 group scored better (11.9%) than the others whereas no participants showed low levels in SAS in the 12–17 group. However, these differences between groups were not statistically significant (χ^2^ = 10.22; *p* > 0.05).

Finally, regarding self-esteem levels, the 12–17 group showed the worst scores, with 30.8% low self-esteem levels, while the >18 group showed only 5.2%. Finally, the 0–5 group showed the best results in high self-esteem (19%), while no participants showed high self-esteem levels in group 6–11. However, in this case, these differences were not statistically significant (χ^2^ = 11.59; *p* > 0.05) (see Table 5).

## 4. Discussion

Considering the total sample, these results suggest that deafness does not significantly affect satisfaction and self-esteem, while it could affect social anxiety. In particular, most of the sample reported high life satisfaction levels and average self-esteem. These data seem to be counterintuitive and unexpected, but are consistent with another study on an Italian sample of deaf people who reported higher levels of satisfaction than a sample of hearing people [36]. Moreover, this could also be interpreted by using some coping strategies or a sort of social desirability linked to the need to show themselves as people with great resiliency who do not complain about their condition, as indicated by the prevalence of average self-esteem scores.

The situation was different with social anxiety: more than a third of the sample showed high social anxiety scores, indicating how deafness could affect more social dimensions than the others. This effect could be related to difficulties in communication that people who are deaf could have in a hearing people context. These difficulties were identified by four studies and are related to skills in terms of communication, language, and a lack of information starting from childhood [29,37,38]. However, external and cultural factors also play a fundamental role as the Italian context is still not inclusive enough toward people who are deaf. For example, in a hospital, they would probably have some difficulties in understanding a diagnosis from a doctor, and so would probably try to understand them labially or would ask friends or family to interpret for them as well as at the post office or other public services [39]. Therefore, in such a context, people who are deaf could be more likely to experience social anxiety.

Regarding the demographic factors, the results of the group comparison by age and gender showed that they did not seem to affect any of the three variables significantly. Regarding the role of ISL, contrary to the hypotheses, the results suggest that it is not a strong protective factor, or at least not entirely. In particular, comparing those who had never learned ISL and those who had learned, we found no significant differences in the satisfaction and self-esteem dimensions. It was expected that people with bilingual education got along well in both the deaf and hearing communities. For that reason, they could tend to have higher self-esteem [40]. This study did not satisfy this expectancy, which showed only a non-significant tendency in the group of those who never learned ISL to have lower self-esteem. The situation is different for social anxiety because knowing sign language seems protective and prevents high social anxiety levels. The results suggest that those who have never learned ISL showed significantly higher anxiety scores than those who had learned it.

These differences can be explained by the fact that ISL can be considered as one more adjustment resource to be more resilient and to make contact with others. It could permit having more instruments for more accessible and more effective communication [41], at least with other signers, which could assist in expressing a lower level of social anxiety, if not within the whole community, then at least within the deaf community.

In contrast, those who have never learned ISL, will have to implement a greater effort to integrate themselves into the hearing community without the possibility of experiencing the facilitator environment represented by people who are deaf signers. For this reason, they could experience higher social anxiety levels [42].

Moreover, in contrast with the second hypothesis, those who learned ISL as a primary language (0–5) did not show the best results in social anxiety, but those who studied ISL in primary childhood (6–11), aside from oral language, did. In particular, the results shown by the 6–11 group (highest SWLS, lowest SAS) as well as those shown by the 12–17 group (lowest SWLS, highest SAS, and lowest RSES) could be considered as a tendency and not as a representation of significant differences between groups. Finally, it can be stated that, in the present sample, deafness did not seem to significantly affect the QoL expressed by satisfaction and self-esteem and only partially affected the social anxiety dimensions. This effect on social anxiety scores only becomes significant when considering knowing ISL as a variable; in terms of those who have never learned ISL, scores were significantly higher in anxiety regardless of the learning time.

In conclusion, these results could be included in a broad social and cultural perspective that includes the relationship between people who are deaf and hearing individuals, in which the use of sign language interpretation could be fundamental. For instance, in a study by Jones et al. [43], participants reported feeling more included and valued when sign language interpretation was provided at social events, and additionally, hearing people who had previously felt uncomfortable interacting with people who were deaf reported feeling more confident and at ease when they were able to communicate through a sign language interpreter. Furthermore, another important factor in building relationships between the deaf and hearing communities is education. Providing education on deaf culture and language can help to reduce misunderstandings and promote mutual understanding and respect [44].

## 5. Limitations

This study had several limitations that should be considered when interpreting our results. These include the overall sample size, which was relatively small, and in particular, the discrepancy between different subgroups in size and the higher prevalence of female participants (70.3%). Moreover, another important limitation was the lack of sociodemographic information on the participants such as education, marital status, the age at which they received the diagnosis, and additional information about their parents (e.g., if they were signers or not).

Additionally, the online recruitment procedures could constitute a limitation: it may naturally select more active individuals on both the Internet and social media platforms, thus neglecting potential individual differences in how people deal with their condition.

Finally, a probable cause of the lack of significance of the tendencies that emerged from the descriptive results was the low sample size due to the size of the population we surveyed.

Future studies should carefully consider these elements, together with a more heterogeneous sample selection, particularly in terms of gender and signers–no signers. Thus, the absence of results should not be treated as definitive; more studies with larger samples are needed to explore and deepen the effect of ISL in the future.

## 6. Conclusions

In conclusion, knowing ISL seems to be not enough and can’t be considered a universal protective factor for QoL of all people who are deaf. These results show that it is surely good for them to know sign language, and it can be helpful to feel more competent and less anxious in communication with other signers, but it is not possible to demonstrate that this is an effective protective factor for other dimensions of quality of life, especially regarding social and everyday aspects within Italian society. 

## Figures and Tables

**Table 1 healthcare-11-01021-t001:** Distribution of gender, age, birth deaf condition and deaf parents in the groups.

ISL					
		>18	<18	Never	Total Sample
		N (%)	N (%)	N (%)	N (%)
Gender	F	34 (58.6)	53 (75.7)	41 (76.0)	128 (75.7)
	M	24 (41.4)	17 (24.3)	13 (24.0)	54 (24.3)
Age	18–30	13 (22.4)	27 (38.6)	17 (31.5)	57 (31.3)
	31–50	37 (63.8)	32 (45.7)	29 (53.7)	98 (53.8)
	>50	8 (13.8)	11 (15.7)	8 (14.8)	27 (14.8)
Birth	Yes	34 (58.6)	59 (84.3)	31 (57.4)	124 (68.0)
	No	24 (41.4)	11 (15.7)	23 (42.6)	58 (32.0)
Parents	Yes	1 (1.7)	34 (48.6)	3 (5.6)	38 (21.0)
	No	57 (98.3)	36 (51.4)	51 (51.0)	144 (79.0)

Note: Information about gender, age, deaf condition at birth, and parents who are deaf in the groups of those who have learned ISL (after and before 18 years) and those who never learned it.

**Table 2 healthcare-11-01021-t002:** Overall means (SD) by gender, age, and *F* statistics of ANOVA between groups.

	Gender			Age		Total Sample
	F	M	18–30	31–50	>50	
SWLS	23.86 (±6.16)	23.67 (±6.94)	24.44 (±5.87)	23.88 (±6.53)	22.19 (±6.82)	23.80 (±6.38)
ANOVA	*F* = 0.34		*F* = 1.15			
SAS	85.08 (±19.69)	81.41 (±18.25)	86.18 (±18.90)	82.88 (±19.90)	83.41 (±18.22)	83.99 (±19.30)
ANOVA	*F* = 1.37		*F*= 0.53			
RSES	19.87 (±4.58)	19.48 (±4.73)	19.09 (±4.29)	19.97 (±4.56)	20.41 (±5.35)	19.76 (±4.62)
ANOVA	*F* = 0.27		*F* = 0.96			

Note: SWLS = Satisfaction with Life Scale; SAS = Social Anxiety Scale; RSES: Rosenberg Self-Esteem Scale.

**Table 3 healthcare-11-01021-t003:** Mean score s(SD) and *F* statistics of ANOVA between groups.

ISL			
	>18	<18	Never
SWLS	23.95 (±5.85)	24.20 (±6.81)	23.13 (±6.41)
ANOVA	*F =* 0.44		
SAS	82.19 (±20.08)	83.87 (±20.11)	86.07 (±21.10)
ANOVA	*F* = 0.56		
RSES	20.66 (±4.03)	19.36 (±4.39)	19.31 (±5.39)
ANOVA	*F* = 1.61		

**Table 4 healthcare-11-01021-t004:** The differences in the SSWLS, SAS, and RSES levels between those who learned ISL and who did not.

		ISL		
		No	Yes	Total Sample
SWLS	Average	33.3%	32.8%	33%
	High	53.7%	57.8%	56.6%
	Low	13.0%	9.4%	10.4%
χ^2^		0.58		
SAS	Average	44.4%	64.1%	58.2%
	High	44.4%	28.9%	33.5%
	Low	11.2%	7.0%	8.2%
χ^2^		6.01 *		
RSES	Average	70.4%	76.6%	74.8%
	High	13.0%	12.5%	12.6%
	Low	16.6%	10.9%	12.6%
χ^2^		1.18		

Note: * *p* < 0.05.

**Table 5 healthcare-11-01021-t005:** The differences in the SWLS, SAS, and RSES levels in the subgroups of those who learnt ISL.

			<18		
		>18	0–5	6–11	12–17
SWLS	Average	36.2%	31%	26.7%	30.8%
	High	56.9%	59.5%	66.7%	46.2%
	Low	6.9%	9.5%	6.7%	23.1%
χ^2^		4.35			
SAS	Average	68.9%	69.%	66.7%	53.8%
	High	25.9%	28.6%	26.7%	6.7%
	Low	5.2%	11.9%	6.7%	0%
χ^2^		10.22			
RSES	Average	82.8%	69%	86.7%	61.5%
	High	12.1%	19%	0%	7.7%
	Low	5.2%	12%	13.3%	30.8%
χ^2^		11.59			

## Data Availability

Data are available upon request due to privacy restrictions.

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
