# Peer review of "The Relationship between Knowing Sign Language and Quality of Life among Italian People Who Are Deaf: A Cross-Sectional Study"

_healthcare, 2023, doi:10.3390/healthcare11071021_

Round 1
Reviewer 1 Report
Interesting manuscript. First off, te authors must use first person language. For example, not adult deaf sujects, but adult subjects who are deaf. This needs to be edited throughout the entire manuscript. Second sentence in introduction says "deafness has to be cured" -- this is not appropriate nor is it possible. Words in parentheses should have the word "and" before the last two words. Much of this manuscript needs editing for spelling, noun-verb agreement, punctuation, tense, and coherence. More detail about subjects needs to be included. What was the breakdown of the 182 participants? This is important. How many items on the questionnaire? You divided the participants into three groups -- was there a difference in the composition of these groups? To obtain these participants what was done to obtain these participants (for each of the three groups)? A table of the decriptive statistics of the three groups would add to the manuscript. You looked at gender, what about age? There should not be one sentence paragraphs. In regards to the Tables -- you might consider reorganizing the manuscript to address all of the information related to Table 4 together and do the same for the other tables. It is difficult to go back and forth. I think there are other limitations that are not included.
Author Response
Dear Reviewer 1,
Thank you for your time and for providing me your helpful comments. I tried to respond to each, you can find the changes highlighted in yellow in the text
1.“the authors must use first person language. For example, not adult deaf subjects, but adult subjects who are deaf. This needs to be edited throughout the entire manuscript”
Response: Thank you for your comment. I edited the manuscript following your indications.
2.” Second sentence in introduction says "deafness has to be cured" -- this is not appropriate nor is it possible.
Response: Thank you for pointing this out. I have modified the sentence. Line 28.
3."Words in parentheses should have the word "and" before the last two words".
Response: Thank you for the comment, I fix it in the manuscript.
4. “Much of this manuscript needs editing for spelling, noun-verb agreement, punctuation, tense, and coherence”.
Response: This revised version was edited by a native English speaker, and I hope the manuscript is more clear and correct now.
5.“More detail about subjects needs to be included”.
Response: Thank you for your indications. I added more information in the participant paragraph. Line 154-158.
6.“What was the breakdown of the 182 participants? This is important”
Response: Thank you for this comment. I have added a specific table and more information about the breakdown of the participants, including more detailed information about the criteria for dividing groups. Line 154-158 and table 1.
7.How many items on the questionnaire?
Response: Thank you for this comment. I added the number of items for each questionnaire in the measures paragraph.
8-9. “You divided the participants into three groups -- was there a difference in the composition of these groups?”; “To obtain these participants what was done to obtain these participants (for each of the three groups)”?
Response: Thank you for pointing this out. First, we recruited all participants, and then we divided them based only on the criterion of knowing ISL or not while for those who knew ISL the criterion was the age in which they learned it. There are no significant differences in the group composition for gender and age (p<0.05) while there are differences in their numerosity I added this information in the text. Line 194-197.
10.“A table of the decriptive statistics of the three groups would add to the manuscript”.
Response: Thank you for your comment. I added a table with descriptive statistics of the sample. Table1-2.
11.“You looked at gender, what about age”?
Response: Thank you for this comment. I divided the sample in three age ranges (18-30; 31-50; >50), and there are no significant differences between them in the mean score of the variables ( p>0.05). I added these information in the results paragraph. Line 223-224.
12.“ you might consider reorganizing the manuscript to address all of the information related to Table 4 together and do the same for the other tables. It is difficult to go back and forth”
Response: Thank you for this comment. I rewrote the results section as you suggested, I hope that the read now is more clear.
13.“I think there are other limitations that are not included”.
Response: Thank you for your comment, I added some others limitations in the paragraph. Line 379-385.
For more details please see the revised version manuscript.
Reviewer 2 Report
This is an interesting study about the connection between learning ISL and having a high quality of life. The hypotheses are relevant to those working with the deaf community.
There are significant grammar mistakes throughout, such as transition words, punctuation, misspellings, word forms, missing words, word order, etc., especially in the first three pages and in section 4: Discussion. There are also some modifying clauses that obscure meaning. Extensive editing is needed.
Author Response
Dear reviewer 2,
Thank you for the time and effort you have dedicated to read and provide your feedback on my manuscript.
An English native speaker checked this revised version and I hope the manuscript is more clear and correct now.
Reviewer 3 Report
This is an interesting manuscript examining the impact of knowing sign language on quality of life among Italian sign language users. While this is an interesting topic, I do have several comments and concerns.
Title: "...sign language on quality of life of deaf...".
I am not certain that the word "impact" is appropriate. This would be more appropriate if the study utilized an experimental design in which causality can be examined. In the current study only correlations are examined.
ABSTRACT - The aim should come before participants.
INTRODUCTION
Line 56 - would be useful to know percentage of people in deaf community who use sign language
Line 74 - research
Line 82 - close quotation marks
Line 85 - In line with social models of disability and in line with deaf community , I would not use the term "suffer from hearing" rather "have hearing impairments". Many people from deaf community would argue that they do not suffer from their condition.
Line 86 - of the 5% who have deaf parents, were all parents signing?
Line 115 - If it is argued that use of sign language can promote satisfying relationships, why are these not examined as a dependent variable. Further, the introduction should clearly define quality of life and the construct that were selected as the dimensions in the current study.
Line 118 - lower instead of less
I would argue against the use of the word "subject" that is more appropriate for an experimental design and would use "participant" or "individual"
METHODS
A lot of information is missing on participants. It is not clear what information was collected in the demographic section and no results on demographics are provided. For example, how many parents are hearing vs. not hearing. Age of hearing loss? Hearing loss severity? All of these demographics should also be examined in the study.
Line 165 - Erase "to" before "higher scores"
Line 181 - Sample does not need to be capitalized. As well, throughout the entire manuscript some words are capitalized some of the time and not in other times. Try to be consistent about this.
Bottom paragraph of page 4, explain why 18 was used as a cut-point?
RESULTS
Table 1 - Why does the second column have F as well as the third? Please add a note at the bottom of the table with the acronyms that are used. I would argue against using acronyms throughout the results section. It is okay in the tables, but it gets confusing and cumbersome in reading the results.
The results section is overly lengthy and repetitive. For one, many results are presented extensively in the text, i.e. stating lower and higher scores, while at the end of each paragraph the reader learns that the differences are in reality not significant; which make the entire previous paragraph irrelevant. If results are not statistically significant differences, it makes not sense to go into length in depicting the differences. Further, using 3.1 the authors highlight the differences (or insignificant differences) using means, the following sections (if I understood this correctly) show pretty much the same findings, only rather than using means, showing them using division into groups. This is redundant. I would choose one of the methods that works best for the authors and only present once.
DISCUSSION
English editing would be useful, especially in the discussion section.
Line 282 - wording "data about could"
Line 300 - belongs to results section and was not shown in the results section
Line 303 - wording cumbersome
Line 307 - As one example of self-esteem capitalized in one sentence and not capitalized in another
Line 318 - wording cumbersome "contrary, who have never" add "individuals who have never"
I would appreciate if the discussion section could tie into the meaning of the current findings to the deaf community versus relationships with people outside the deaf community
Author Response
Dear reviewer 3,
Thank you for your time and effort in reading my manuscript and providing me such detailed comments
Here is a point-by-point response to your comments and concerns. You can find the changes highlighted in yellow in the text
Title: "...sign language on quality of life of deaf...".
I am not certain that the word "impact" is appropriate. This would be more appropriate if the study utilized an experimental design in which causality can be examined. In the current study only correlations are examined.
Response: I modified the title in :” The relationship between knowing Signs Language and Quality of Life among Italian deaf people: A Cross- Sectional Study” according to the point you raised
ABSTRACT
- “The aim should come before participants”.
Response: Thank you for the comment, I modified the text following your suggestion. Line 15-18
INTRODUCTION
- “Line 56 - would be useful to know percentage of people in deaf community who use sign language”
Response: Thank you for the comment. I did not find clear data from official sources, for this reason, i did not added this information in the text.
- Line 74 – research
Response: Thank you, I modified it according to your observation.
- “Line 82 - close quotation marks”
Response: Thank you, I modified this refusal according to your observation.
- “Line 85 - In line with social models of disability and in line with deaf community, I would not use the term "suffer from hearing" rather "have hearing impairments". Many people from deaf community would argue that they do not suffer from their condition.”
Response: Thank you for your observation. I agree with that, I modified the text according to that.
- Line 86 - of the 5% who have deaf parents, were all parents signing?
Response: Thank you for the question. The 5% refers only to deaf parents who are also signers. I modified the text to make it more clear Line 85-88.
- “Line 115 – “If it is argued that use of sign language can promote satisfying relationships, why are these not examined as a dependent variable. Further, the introduction should clearly define the quality of life and the construct that were selected as the dimensions in the current study”
Response: Thank you for this interesting observation. We choose to focus on general life satisfaction (which includes also relationships in a certain way), and two psychological variables that we think could be central in this kind of disability from an “individual” point of view. Your observation is very useful in suggesting us ideas for further studies by which we can explore the relational aspects of life widely. Thank you very much.
- “Line 118 - lower instead of less”
Response: Thank you for pointing this out. I fixed it in the text.
- “I would argue against the use of the word "subject" that is more appropriate for an experimental design and would use "participant" or "individual"
Response: Thank you for this comment, I fixed it in the manuscript.
METHODS
- “A lot of information is missing on participants. It is not clear what information was collected in the demographic section and no results on demographics are provided. For example, how many parents are hearing vs. not hearing. Age of hearing loss? Hearing loss severity? All of these demographics should also be examined in the study”.
Response: Thank you very much for this interesting point. I have included the information you ask in the paper, in the participant’s section and in a specific table (table 1) regardless of the age of hearing loss because unfortunately, we did not have this information (you can find it as a limitation in the final paragraph)
- “Line 181 - Sample does not need to be capitalized. As well, throughout the entire manuscript some words are capitalized some of the time and not in other times. Try to be consistent about this.”
Response. Thank you for your observation. I fixed that all along the manuscript.
- “Bottom paragraph of page 4, explain why 18 was used as a cut-point”?
Response: Thank you for the question. We choose to divide groups following developmental age in which they learn ISL (early childhood, late childhood, adolescence) in which primary and secondary education could help to integrate Italian sign language with the communication skills in the development, and we use 18 as cut- off point as the end of adolescence (and primary and secondary education in Italy) and the beginning of early adulthood.
RESULTS
- “Table 1 - Why does the second column have F as well as the third? Please add a note at the bottom of the table with the acronyms that are used. I would argue against using acronyms throughout the results section. It is okay in the tables, but it gets confusing and cumbersome in reading the results”
Response: Thank you for your question. It was F coefficient of ANOVA but it was unclear in that position. I reorganized the tables in the manuscript and I hope now they are more clear to the reader.
- “The results section is overly lengthy and repetitive. For one, many results are presented extensively in the text, i.e. stating lower and higher scores, while at the end of each paragraph the reader learns that the differences are in reality not significant; which make the entire previous paragraph irrelevant. If results are not statistically significant differences, it makes not sense to go into length in depicting the differences. Further, using 3.1 the authors highlight the differences (or insignificant differences) using means, the following sections (if I understood this correctly) show pretty much the same findings, only rather than using means, showing them using division into groups. This is redundant. I would choose one of the methods that works best for the authors and only present once”.
Response: Thank you for this useful comment. I rewrote the results section, summarizing some parts and focusing mainly on the levels and not on the means differences. I hope this part is more clear now.
DISCUSSION
- “English editing would be useful, especially in the discussion section”.
Response: Thank you for your comment. I have edited the manuscript with the help of an native English speaker and I addressed the point you raised in that section hope the manuscript is correct now.
- “I would appreciate itif the discussion section could tie into the meaning of the current findings to the deaf community versus relationships with people outside the deaf community”
Response: Thank you for the comment. I added a part in the discussion paragraph to address your suggestion.
For more details please see the revised version manuscript.
Round 2
Reviewer 1 Report
Many grammatical errors still present in wording. In fact, the title uses Signs Language which is incorrect. Also there are problems with tense throughout. Some of the same things I said before. Author do not use person first language (i.e., people who are deaf vs. deaf people). Demographic information on participants is still incomplete. This study could not be replicated. Results -- what were the differences by the different groups? Isn't this the point of the study? The other results are better at helping to explain the study. Good addition to discussion and limitations.
Author Response
Dear Reviewer 1,
Thank you for answer us so quickly and for your comments; here it s a point-to-point response hoping to address the point you raised:
1."Many grammatical errors still present in wording. In fact, the title uses Signs Language which is incorrect; also there are problems with tense throughout. Some of the same things I said before."
Response: Thank you for pointing this out; now I have fixed the errors left in the manuscript
2."Author do not use person first language (i.e., people who are deaf vs. deaf people)".
Response: Thank you for pointing this out; now I have fixed the errors to make a more coherent in the manuscript
3."Demographic information on participants is still incomplete. This study could not be replicated".
Response: Thank you for your comment. I agree with you about the lack of demographic information, we focused only on information more related to deafness. Unfortunately, we reported all the information we have at our disposal. However, we have underlined how, the lack of additional sociodemographic information is an important limitation of the study in the last paragraph. I hope it is enough to address this problem.
4) "Results -- what were the differences by the different groups? Isn't this the point of the study? The other results are better at helping to explain the study".
Response. Thank you for your comment. We rewrote the results part to respond to one of the core concerns that reviewer 3 raised to us. He/ she asked us to reduce the redundancy of this part and to choose only one method to report results that overlapped in his/her perspective. For that reason, we had to summarize the part in which means were reported emphasizing only the group differences by levels of the variables using cut-off. In our opinion, this method could describe differences in terms of risk level, which is more informative and significant for the study purposes.
Reviewer 3 Report
Authors have improved the manuscript in line with my previous comments.
Author Response
Dear Reviewer 3,
Thank you for your quick answer and your interest towards our manuscript.